# Spontaneous urinary bladder regeneration after subtotal cystectomy increases YAP/ WWTR1 signaling and downstream BDNF expression: Implications for smooth muscle injury responses

**Karen J. Aitken**[1,2]*, **Priyank Yadav**[1,3,4ʘ], **Martin Sidler**[1,2,4,5ʘ],
**Thenuka Thanabalasingam**[1,6], **Tabina Ahmed**[1,6], **Prateek Aggarwal**[1,2,6], **Shing Tai Yip**[1],
**Nefateri Jeffrey**[1], **Jia-Xin Jiang**[1,7], **Aliza Siebenaller**[1], **Chris Sotiropoulos**[1,6,7], **Ryan Huang**[6],
**David Minh Quynh Le**[6], **Paul Delgado-Olguin**[8,9], **Darius Bagli**[1,2,4,7]*

1 Developmental and Stem Cell Biology, Research Institute, Hospital for Sick Children, Toronto, Ontario, Canada, 2 Institute of Medical Sciences, Faculty of Medicine, University of Toronto, Toronto, Ontario, Canada, 3 Department of Urology and Renal Transplantation, Sanjay Gandhi Postgraduate Institute of Medical Sciences, Lucknow, Uttar Pradesh, India, 4 Urology Division, Department of Surgery, Hospital for Sick Children, Toronto, Ontario, Canada, 5 Division Chief, Paediatric and Neonatal Surgeon, University Hospital Ulm, Ulm, Baden-Württemberg, Germany, 6 Human Biology Programme, Faculty of Arts and Sciences, University of Toronto, Toronto, Ontario, Canada, 7 Department of Physiology, Faculty of Medicine, University of Toronto, Toronto, Ontario, Canada, 8 Translational Medicine Program, Research Institute, Hospital for Sick Children, Toronto, Ontario, Canada, 9 Department of Molecular Genetics, Faculty of Medicine, University of Toronto, Toronto, Ontario, Canada

ʘ These authors contributed equally to this work.
* karen.aitken@alumni.utoronto.ca (KJA); darius.bagli@sickkids.ca (DB)

## Abstract

Rodents have the capacity for spontaneous bladder regeneration and bladder smooth muscle cell (BSMC) migration following a subtotal cystectomy (STC). YAP/WWTR1 and BDNF (Brain-derived neurotrophic factor) play crucial roles in development and regeneration. During partial bladder outlet obstruction (PBO), excessive YAP/WWTR1 signaling and *BDNF* expression increases BSMC hypertrophy and dysfunction. YAP/WWTR1 and expression of *BDNF* and *CYR61* were examined in models of regeneration and wound repair. Live cell microscopy was utilized in an *ex vivo* model of STC to visualize cell movement and division. In Sprague-Dawley female rats, STC was performed by resection of the bladder dome sparing the trigone, followed by closure of the bladder. Smooth muscle migration and downstream effects on signaling and expression were also examined after scratch wound of BSMC with inhibitors of YAP and BDNF signaling. Sham, PBO and incision (cystotomy) were comparators for the STC model. Scratch wound *in vitro* increased SMC migration and expression of *BDNF*, *CTGF* and *CYR61* in a YAP/WWTR1-dependent manner. Inhibition of YAP/WWTR1 and BDNF signaling reduced scratch-induced migration. *BDNF* and *CYR61* expression was elevated during STC and PBO. STC induces discrete genes associated with endogenous *de novo* cell regeneration downstream of YAP/WWTR1 activation.

**Data Availability Statement:** Data are available at https://github.com/kjaitken/STC_repository.

**Funding:** This work was supported in part by the Strategic Training Program in Regenerative Medicine - CIHR (KJA) and the 2022 Urology Care Foundation Research Scholar Award Program and the AUA Northeastern Section (PY). The funders had no role in study design, data collection and analysis, decision to publish, or preparation of the manuscript, and only played a role in support of the scientific training of the fellow KJA in regenerative medicine and PY in urology research. No additional external funding was received for this study.

**Competing interests:** The authors have declared that no competing interests exist.

**Abbreviations:** BDNF, Brain-derived neurotrophic factor; BSMC, bladder smooth muscle cell; CNN1, Calponin 1; CTGF, Connective tissue growth factor; Cyr61, Cysteine-rich angiogenic inducer 61; mTOR, mammalian target of rapamycin; NTRK2 or TRKB T1, Neurotrophic tyrosine receptor kinase (BDNF receptor); PBO, partial bladder outlet obstruction; SMC, smooth muscle cell; STC, subtotal cystectomy; VP, verteporfin; WWTR1, Transcriptional co-activator with a PDZ-binding [WW] domain containing transcription regulator 1; YAP, Yes-associated protein.

# Introduction

Regeneration is a critical process prevalent among all animal kingdoms and provides the capacity for repair, regrowth and replacement of old, diseased or damaged cells, tissues and organs [1, 2]. However, the potential for regeneration varies amidst species, organs and age-groups [3–5]. In 1928 and 1939, early work demonstrated that the bladder regenerates in rabbits and humans after subtotal cystectomy (STC) [6, 7]. This regeneration occurs in the absence of scaffolds or molds and patients regain bladder capacity of up to 410 mL [8–10]. In interstitial cystitis patients, regeneration depends upon the region resected. Bladders subjected to subtrigonal resection, with ureters re-implanted anteriorly, do not recover structure or function [11]. In contrast, bladders subjected to supra-trigonal resection (i.e., essentially the same as STC) do recover.

Understanding the processes controlling bladder regeneration is required to engineer functional tissues suitable for restorative therapy [12]. However, efforts to replace damaged tissue in the bladder have often focused on combining exogenous stem or somatic cells with tissue scaffolds. In 2006, Atala *et al*, presented a seminal clinical study in which neuropathic bladders from myelomeningocele patients were transplanted with collagen-polyglycolic acid scaffolds seeded with autologous smooth muscle cells (SMC) [13]. Though the implanted cell-seeded scaffolds increased bladder capacity, these bladders lacked spontaneous contraction ability suggesting that restoration of SMC phenotype and innervation may be required to improve bladder function.

Promoting endogenous regeneration *in situ* could be utilized for benign conditions, and is distinct from bladder replacement that employs biomaterials and scaffolds to increase capacity, with or without cell-seeding [12]. Rodent and rabbit bladders return almost completely to their original size, normal morphological muscle arrangement and re-innervation after STC [14, 15]. In the rat bladder, this process is accompanied by upregulation of developmental signaling pathways including those controlled by hedgehog/GLI Family Zinc Finger proteins (SHH/GLI1,2,3), and bone morphogenic proteins (BMPs) [16, 17]. Growth of SMC have been shown to increase with subtotal cystectomy [15], and SMC migration occurs during wound healing [16, 18]. Despite this, regulators of bladder regeneration and migration of bladder SMC remain poorly understood. Mechanistic insight into the role of additional developmental pathways will undoubtedly be crucial to advancing molecular approaches that support endogenous bladder regenerative capacity for therapeutic benefit.

Yes-associated protein (YAP) and Transcriptional co-activator with a PDZ-binding [WW] domain containing transcription regulator 1 (WWTR1) are transcriptional coregulators that act as signaling effectors initiated by inactivation of the protein kinase Hippo/Mst1. This conserved signalling cascade controls cell proliferation, apoptosis and development in many organs [19–21]. YAP/WWTR1 is activated in response to mechanical stress [22–25]. Signaling through YAP/WWTR1 in the bladder leads to upregulation of brain-derived neurotrophic factor (BDNF), connective tissue growth factor (CTGF/CCN2) and cysteine-rich angiogenic inducer 61 (CYR61/CCN1) [26]. The latter two are part of the "CCN" matricellular group of proteins [27]. CCN (cellular communication network factor) proteins are linked to collagen production, proliferation and bladder obstructive responses [28, 29], while YAP/WWTR1 are associated with hypertrophic growth in bladder and other muscle cells [26, 30]. Further, YAP/WWTR1 appear to be required for bladder development and growth, as knockouts reveal roles in both kidney and bladder development [31, 32]. However, the loss of urine production in the YAP/WWTR1 knockout kidney may reduce the physiologic mechanical stimulation of the bladder and downstream signaling for growth. The activity of YAP and WWTR1 during endogenous regeneration in the bladder has been unexplored. Here, we interrogated the role

of the YAP/WWTR1 pathway and downstream BDNF, CTGF and CYR61 in the regulation of bladder regeneration, as compared to wound healing and obstructive myopathy *in vivo*, and in smooth muscle cell (SMC) migration *in vitro* and *ex vivo*.

## Materials and methods

### Bladder smooth muscle cell scratch wound assay

Human bladder smooth muscle cells (BSMC, Sciencell, Carlsbad, California, USA) were maintained in SMC media (Sciencell). BSMC were plated at a density of $4x10^4$ cells/mL in EMEM (Wisent, St-Bruno, Quebec, CANADA) containing 10% heat-inactivated Fetal Calf Serum (FCS, Wisent). Once confluent, BSMC were serum-starved for 24 hours, then scratched with a 10 μL micropipette tip or left unscratched. Cells in both groups were subsequently washed with warm PBS three times and incubated in starvation medium with 0.1% FCS. Cells were incubated for 2.5 hours for RNA extractions, and for 18 hours for live cell microscopy and immunocytochemistry. In the RNA extraction studies, cell monolayers on 100 mM dishes were left either unscathed or scratch wounded in three parallel lines, in two directions, using a multichannel pipette set with three 10 μL pipette tips. Cells were treated with vehicle, Verteporfin (0.1 μM, Sigma, St. Louis, Missouri, USA) or soluble chimaeric NTRK2-Fc receptor (0.1μg/ml, Alamone Labs, Jerusalem, Israel).

### Live cell microscopy of migration in scratch wound and ex vivo bladder

Scratch wounds were applied to BSMC monolayers grown in coated 96 well glass plates. Visualization of BSMC migration was performed by light microscopy with Volocity 6.3 software (Perkin Elmer, Waltham, Massachusetts, United States) over 18 hours to determine the percent growth into the scratch. For the *ex vivo* bladders, 12 week old mice were harvested under anaesthesia and bladders harvested into DMEM plus antibiotic/antimycotic (Wisent). The smaller size of mouse bladders enables fluorescent visualization of cell migration in the muscle and STC region under the confocal microscope. Bladders were either unmanipulated or underwent an ***ex vivo*** STC, where 50–75% of the bladder was removed and the remaining bladder sutured in a running pattern. *Ex vivo* bladders were treated with Carboxyfluorescein Diacetate Succinimidyl Ester (CFSE) for 30 minutes in warm PBS, then washed in PBS, and media re-added. As an alternate method of fluorescent tracing to enhance SMC visualization, AAV6-GFP were injected into the bladder wall in 6 regions. Multiple regions of the bladder were selected for tracking migration overnight. In some cases, z-stacks were also performed, but the number of regions were decreased in these cases, due to the large size of the ensuing files that can cause the program to cease functioning.

### Spontaneous regeneration and obstruction in the rat

Six-week old female Sprague-Dawley rats were randomized to 1 or 7 weeks for sham, STC and incision control (i.e. cystotomy). Anesthesia was administered to rats by Isofluorane at a 2.5 mL flow rate with 3% $O_2$. The Animal Care and Use Committee (ACC) of the Hospital for Sick Children has approved this study. Isofluorane was utilized for anesthesia and long-lasting Temgesic (0.1 mg/kg) utilized for pain management in the 3 days post-surgery, and then as needed. To alleviate suffering, the number of rats were minimized, and analgesia applied where needed. After the bladder was exposed, 2 sidestays were placed on the bladder and 75% of the bladder was removed including the dome sparing the trigone and the vesicoureteral junctions. The remaining bladder was sutured in a running pattern with 8.0 silk, then stays removed. The fascia and abdominal wall were closed, and the animal brought out of anesthesia

on a recovery bed with heat and supplemental oxygen. To control for the incision and wound healing, an incision was created on the ventral side, then sutured without bladder removal. This incision control is distinct from partial loss of bladder tissue during STC as the size of the bladder remains the same. To provide another comparison to STC, we performed partial bladder outlet obstruction (PBO) which induces hypertrophic myopathy. PBO for 7-weeks was performed by ligating around the proximal urethra and a 0.9 mm rod, which was then removed, as in Sidler *et al*, 2018 [26]. At the end of the timecourse, each animal was sacrificed by exsanguination according to the standard of protocol set by the ACC. While rats with STC or incision recovered well and did not show signs of distress, some animals with PBO required early sacrifice to alleviate suffering. Residual volumes were measured by collection of urine at the time of bladder harvest during anesthesia induction in the rats as performed previously [26].

## Immunofluorescent staining

Cells from culture were fixed in 4% PFA for 15 minutes at room temperature. Cryosections from sham, incision control and STC or BSMC were fixed in 4% PFA for 20 minutes at RT, then stopped in 0.1 M Glycine in PBS. After washing, the samples were then incubated with 0.2% Triton X-100 for 10 minutes at RT, and subsequently washed 3 times in PBS. Nonspecific sites were blocked with 10% Donkey Serum + 10% Bovine Serum Albumin for 1 hour, room temperature. Primary antibodies (YAP/WWTR1, 1:100—Santa Cruz Biotechnology, Inc., Dallas, Texas, USA, beta-3 tubulin, 1:3000—abcam, calponin and alpha-smooth muscle actin, 1:200—Sigma) or normal IgG dilution in block, were incubated on sections or cells at 4˚C overnight, followed by 3 washes in PBS. Secondary antibodies (1:1000 dilution in block) were incubated 1 hour at room temperature. Samples were then incubated with nuclear staining dye (Hoechst), washed with 0.1% Triton X-100 in PBS and mounted in Dako Fluorescence mounting medium. For YAP/WWTR1, methanol was used to pre-clear samples prior to the blocking step [26, 33]. Fluorescence was imaged using spinning disk confocal microscopy and intensity of signals measured using Volocity software 6.3.

## Expression analysis by QPCR

RNA extractions and real-time RT-PCR were performed as in Sidler *et al*, 2018 [26]. For RNA isolation from scratch wounds, cells were scraped into Trizol (ThermoFisher, Waltham, Massachusetts, USA). RNA from bladder tissue was isolated by first crushing under liquid nitrogen with a mortar and pestle then homogenizing in Trizol with 0.3–1.5 mm steel pellets, using an ice-cold Bullet Blender (NextAdvance Technologies, Troy, NY, USA) [26, 33, 34]. For both cells and tissue, chloroform was added, and phases were separated by centrifugation. Next, the RNA in the aqueous phase was precipitated in Isopropanol, and washed in 70% ethanol. RNA was resuspended in >40 µL of DEPC-treated water, and then tested on a nanodrop for quality and quantity. 200 ng of RNA was reverse transcribed into cDNA using Superscript III with Oligo dT [10–18] (ThermoFisher). cDNA was PCR amplified on the MJ Research PCR machine running Opticon software, with primers designed to span exons in genes (Integrated DNA Technologies, Coralville, Iowa, USA, Table 1). The delta-delta c(t) method was utilized to analyse relative quantities of target vs. reference genes (including Human UBC, B2M, HPRT, RPL13, rat Hprt, Rpl32, Actb).

## Enzyme-linked immunoassay (ELISA) for BDNF in cultured human cells

Protein lysates from scratch-wounded or control cells +/- Verteporfin (VP) were collected in RIPA buffer plus protease inhibitors, cleared of cell debris by centrifugation and frozen at

**Table 1. QPCR oligonucleotides.**

| Gene name | Forward primer (5'-3') | Reverse primer (5'-3') | Amplification product (bp) |
|---|---|---|---|
| Human and rat CYR61/CCN1 | CGAGGTGGAGTTGACGAGAAA | CTTTGAGCACTGGGACCATGA | 159 |
| Human and rat CTGF/CCN2 | AAGACCTGTGGGATGGGC | TGGTGCAGCCAGAAAGCTC | 194, 193 |
| Human pan-BDNF | TAACGGCGGCAGACAAAAGA | GAAGTATTGCTTCAGTTGGCCT | 101 |
| Human YAP | ACCCTCGTTTTGCCATGAAC | TGTGCTGGGATTGATATTCCGTA | 221 |
| Human WWTR1 (TAZ) | CCATCACTAATAATAGCTCAGATC | GTGATTACAGCCAGGTTAGAAAG | 338 |
| Human B2M | AGGCTATCCAGCGTACTCCA | TCATCCAATCCAAATGCGGC | 348 |
| Human HPRT | GGCGAACCTCTCGGCTTT | CATCACTAATCACGACGCCA | 159 |
| Human, rat HPRT | GATGATGAACCAGGTTATGAC | GTCCTTTTCACCAGCAAGCTTG | 471, 472 |
| Human UBC | TTAGGACGGGACTTGGGTGA | TCACGAAGATCTGCATTGTCAAG | 240 |
| Human ACTB (beta-actin) | CGGCTACAGCTTCACCACCA | CGGGCAGCTCGTAGCTCTTC | 140 |
| Rat pan-Bdnf | GGCTGACACTTTTGAGCACG | GCTGTGACCCACTCGCTAAT | 273 |
| Rat Bdnf isoform exon VI-var1 | AATGTGACTCCACTGCCGG | TCACTCTTCTCACCTGGTGGAA | 69 |
| Rat Bdnf isoform exon IV-var5 | AGCTGCCTTGATGTTTACTTTG | CGCCCATGAAAGAAGCAAAC | 137 |
| Rat Yap | AATATCAATCCCAGCACAGCA | CATTCTGAGTCCCTCCATCC | 109 |
| Rat Wwtr1 (TAZ) | GAAGGTGATGAATCAGCCTCTG | GTTCTGAGTCGGGTGGTTCTG | 169 |
| Rat Ntrk2 var2 (NM_001163168.3) | GAGCATCTCTCGGTCTATGC | CCCATCCAGGGGGATCTTAT | 147 |
| Rat Hprt cDNA | AGGCCAGACTTTGTTGGATT | GCTTTTCCACTTTCGCTGAT | 118 |
| Rat Rpl32 cDNA | CATCTGTTTTGCGGCATCA | CACCCTGTTGTCGATGCCTC | 152 |
| Rat Actb (beta-actin) | GGTCGTACCACTGGCATTGTG | GCTCGGTCAGGATCTTCATGAG | 150 |

-20°C. Standards or 10 ug of protein from lysates were added to sample diluent (1:1) from the sandwich ELISA kit immunoassay kit for BDNF (R&Dsystems, Burlington, Massachusetts, USA), and incubated overnight 4°C. After washing, anti-BDNF, further washes, streptavidin, washes, TMB colour development reagent and STOP solutions were each added in sequence to the wells. The plate was read at 450 nM on a visible plate reader with SoftMax software [26].

## Statistical analysis

Analysis of variance and post-hoc student's t-tests were performed to assay differences between means of each group, with $p < 0.05$ considered significant. Associations between residual volumes and bladder weights were examined by bivariate correlations. Differences in expression were analysed by ANOVA then student's t-tests between linear expression values ($2^{-ddct(t)}$), with $p < 0.05$ considered significant. Where the range of linear expression data was >100-fold (e.g. PBO vs. sham), logarithmic delta c(t) values were utilized for analysis, as indicated.

## Results

### YAP/WWTR1 activity increases in bladder SMC (BSMC) when subjected to migratory stimuli *in vitro*

To assess the function of YAP signalling in bladder SMC to the growth stimulus of a scratch wound assay, we quantified YAP protein in the nucleus of SMC and the levels of mRNA of its downstream targets *Cyr61* and *Ctgf* in human BSMC. Immunofluorescence revealed that YAP/WWTR1 was increased ($p < 0.001$) in nuclei of SMC at the leading edge of the scratch wounded BSMC monolayer, but not distal to the scratch (Fig 1A and 1B). Cells were also treated prior to the scratch wound with verteporfin (VP), a pharmacologic inhibitor that sequesters YAP/WWTR1 in the cytoplasm [35]. Using time lapse micrography, bladder SMC migration was inhibited at the scratch wound edge by VP (Fig 1C, S1 Video).

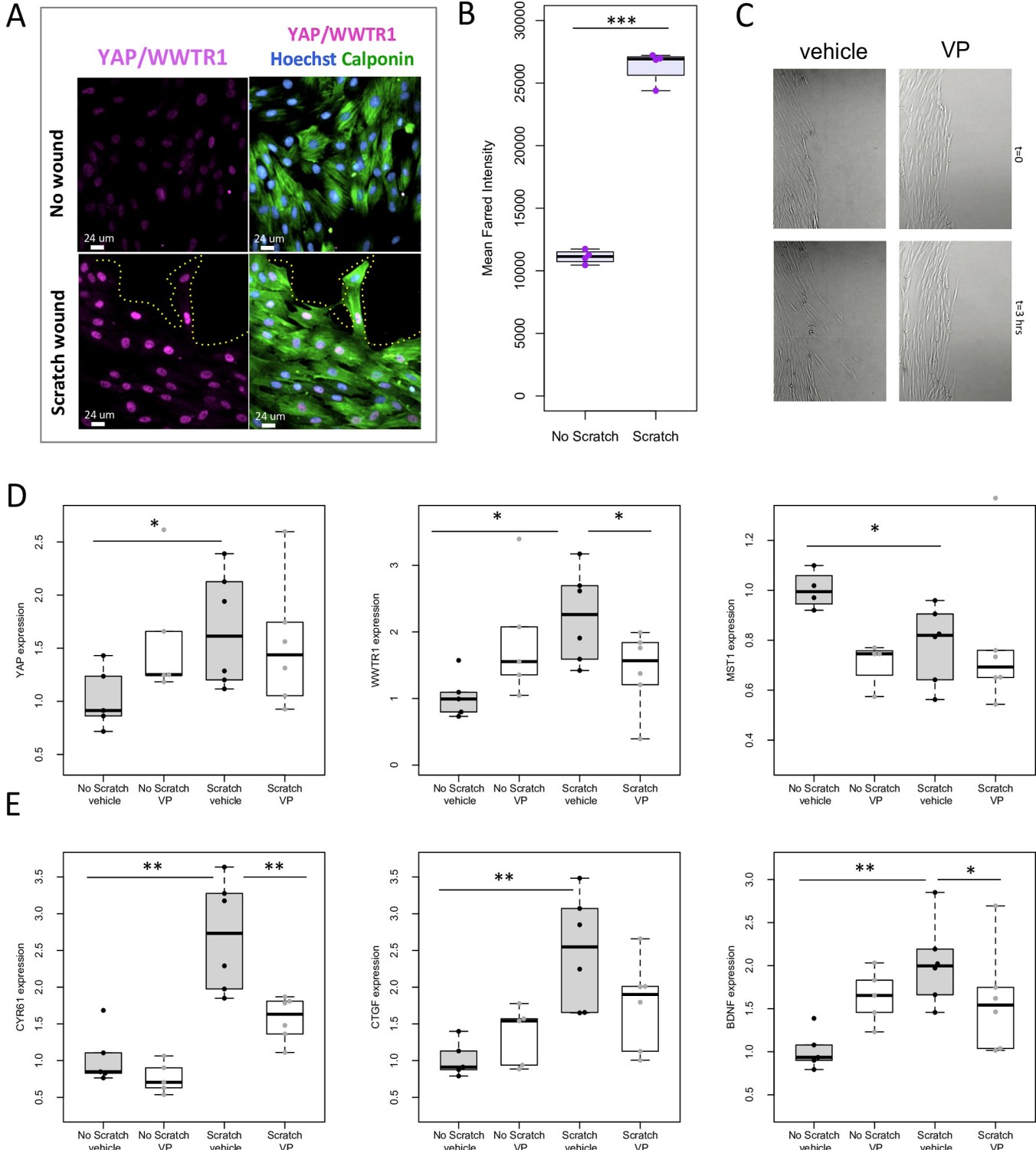

**Fig 1. Wound healing of human bladder smooth muscle cell (BSMC) monolayers increases the regenerative pathway, YAP/WWTR1, and transcription of downstream targets.** As a cognate for wound healing during initial stages of spontaneous regeneration, scratch wounds were created in confluent monolayers of quiescent BSMC. **A, B**: *In vitro* scratch wound induces YAP/WWTR1 signaling and downstream target genes. Scratch wound assay of cultured bladder SMC caused increased intensity of nuclear YAP in migrating cells on the wound edge at 6 hours. n = 4. **C**: Representative images of migrating bladder SMC after scratch wound time-lapse microscopy with and without verteporfin (VP), an inhibitor of YAP/WWTR1 activity (Supporting information **S1 Video** shows time lapse micrography of the SMC migration). **D**: YAP and WWTR1 mRNA expression levels were increased by scratch wound healing. VP lowered the WWTR1 expression induced by wound healing. MST1 expression decreased after scratch wounding. **E**: Scratch wound of BSMC increases expression of

downstream target genes of YAP & WWTR1 signaling, *BDNF*, *CYR61* and *CTGF*. The inhibitor of Verteporfin (VP, 0.1 μM) significantly reduced scratch wound-induced expression of CYR61 and BDNF. *, $p < 0.05$; **, $p < 0.01$; ***, $p < 0.005$.

*Yap* and *Wwtr1* mRNAs increased after scratch wounding (1.8 and 2.2-fold, respectively, $p < 0.05$). VP inhibited scratch-induced expression of *Wwtr1* but not *Yap* (Fig 1D). Interestingly, Mst1, which inhibits Yap/Wwtr1 activity, is transcriptionally downregulated during scratch wounding, but VP induced a further decrease (Fig 1D). The downstream targets of Yap/Wwtr1 in bladder smooth muscle [26], including *Bdnf*, *Cyr61* and *Ctgf* were also examined. *Bdnf* was increased by 2.1-fold in scratched cell cultures as compared with unwounded controls (Fig 1E). Accordingly, qPCR revealed that *Cyr61* and *Ctgf* increased by 2.7 and 2.5-fold ($p < 0.01$), respectively (Fig 1E). Treatment with VP prevented the scratch-induced expression of *Cyr61* and *Bdnf*, with a trend in *Ctgf*.

## *The bladder* regenerates 7 weeks after STC

After incision, bladder mass increased by ~50% at 1 and 7 weeks (Fig 2A). Similarly, bladder to body mass ratio increased after incision by 50 and 27% at 1 and 7 weeks, respectively (Fig 2B). STC reduced the residual volumes of bladders at 1 week to 63% of sham controls (Fig 2C). Bladder mass increased at 1 and 7 weeks after STC above STC at time = 0 levels (dashed line), but did not return to sham levels at this timepoint (Fig 2A). At 7 weeks, residual volumes in STC bladders increased to levels comparable to sham controls. In contrast, residual volumes in incised bladders at 1 week of age increased by 69% above sham controls (Fig 2C). Seven week-old incised bladders had residual volumes decreased by 58% volume compared to 1 week-old incised bladders (Fig 2C). At 7 weeks, residual volumes were 60% lower among incision treatments vs. STC treatments (Fig 2C). Residual volumes in bladders subjected to PBO at 7 weeks significantly increased when compared to sham (Fig 2C). Bladder to body mass ratios were reduced by 26% after 1 week of STC vs shams, but after 7 weeks of STC, ratios were comparable to sham (Fig 2B). In contrast, 1 and 7 weeks after incision repair the body mass ratio increased 48% and 32% above sham, respectively. Thus, residual volumes, bladder mass, bladder/body mass ratio of PBO significantly differed from sham and from STC (Fig 2).

As YAP and WWTR1 activity is regulated post-translationally by phosphorylation and nuclear localization, we next assessed the distribution and levels of YAP and WWTR1 proteins using antibodies that detect each protein. YAP and WWTR1 staining was more prominent after 7 weeks STC (Fig 2D and 2E) in comparison to sham controls. Collagen staining was also examined and showed a change in distribution into a more disorganized pattern (Fig 2D). YAP signal was localized to both nuclei and cytoplasm of smooth muscle cells, revealed by immunofluorescence for calponin (Fig 2E). The nuclear YAP was also detected in both calponin-positive and–negative cells in STC bladders.

## Bladder regeneration is associated with YAP/WWTR1 signaling and BDNF expression

To assess the activity of YAP/WWTR1 during bladder regeneration, we quantified downstream targets *Cyr61*, *Ctgf*, *Bdnf* and *NTRK2* (also known as TRKB T1) by qPCR in bladders after 1 and 7 weeks of STC vs sham operation. Pan *Bdnf* was increased in STC *vs* sham after 1 week (1.68-fold+/-0.53) and 7 weeks (3.6-fold+/-1.1). PBO led to a highly significant increase after 7 weeks (>5-fold, Fig 3A). The total levels of *Bdnf* expression were not affected by incision, comparable to those induced by sham or STC (Fig 3A). While *Bdnf* expression has been studied in the bladder, the response of specific isoforms to regeneration is unknown.

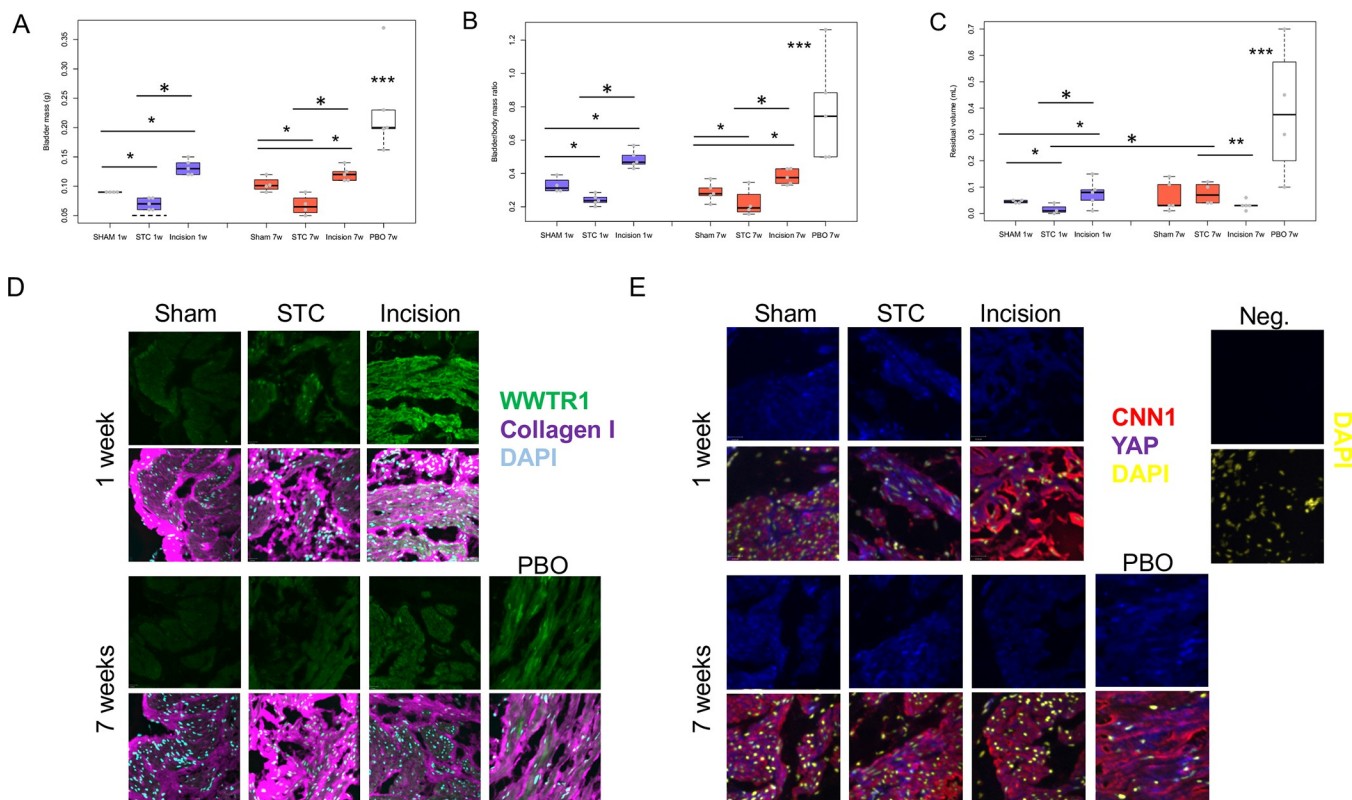

**Fig 2. Spontaneous regeneration occurs rapidly after subtotal cystectomy (STC).** One week post-STC, bladder mass (A) is lower than sham or cystotomy bladders, but STC bladders are higher than t = 0 partial cystectomy bladders (dotted line). Dotted line indicates mean size of STC bladders cut at the same levels as other STC bladders that were removed and weighed on day 0. **A**: After 1+7 weeks, the STC **bladder mass** is lower than sham or cystotomy bladders, but is still higher than the time = 0 STC control (dashed line). **B**: **Bladder to body weight ratios** decreased in STC, while incision of the bladder, as a control for wound healing without cystectomy, lead to increased bladder/body weight ratio at 1 week, but not 7 weeks after incision. **C**: **Residual urine volumes** are significantly less in week1 STC, but regain lost residual volume by week7 STC. **D**: **Immunofluorescent staining for WWTR1 and Collagen I** in sham, STC, cystotomy and PBO treatments, at 1 and 7 weeks. PBO was performed by partial closure of the peri-urethra in Sprague-Dawley female rats. Collagen deposition is dysregulated in both STC and cystotomy treatments, with a significant increase in WWTR1 expression in cystotomy treatments at 1 week. WWTR1 expression at 7 weeks is slightly increased in incision treatments, with an even slighter increase in expression + collagen dysregulation upon PBO which represents SMC hypertrophy. **E**: **Immunofluorescent staining for YAP and Calponin1 (CNN1)** in sham, STC, cystotomy, and PBO treatment, at 1 and 7 weeks. YAP expression is similar at 1 and 7 weeks across all treatments. However, STC treatment at 1 week shows SMC hypertrophy, and even increased CNN1 dysregulation in cystotomy treatments. PBO treatment demonstrates slightly decreased expression of CNN1, with maintained SMC hypertrophy at 7 weeks. *, $p < 0.05$; **, p<0.01; ***, p<0.005.

Therefore, we assessed the expression of *Bdnf* variants 1 and 5, which includes exon IV and VI, respectively, after PBO, STC and incision repair. Variant 1 levels were comparable to sham in response to STC and incision (Fig 3B). However, variant 5 increased by 3.2-fold after 7 weeks STC over sham (Fig 3C). In contrast, both variants 1 and 5 robustly increased >8-fold after PBO as compared to sham (Fig 3B and 3C).

*Cyr61* increased by 1.8- and 2.6-fold in bladders subjected to STC at 1 and 7 weeks, respectively (Fig 3E). Unexpectedly, no significant change in *Ctgf* expression was observed between STC and sham (Fig 3D). Induction of hypertrophic disease by PBO led to increased expression of both *Cyr61* (3.5-fold +/- 0.8) and *Ctgf* (>14-fold) (Fig 3D and 3E). In contrast, Incision alone increased neither *Cyr61* nor *Ctgf* expression (Fig 3D and 3E). However, compared to STC, incision was significantly reduced in *Ctgf* expression at 7 weeks (Fig 3D).

While the BDNF receptor, NTRK2, is known to be induced by mechanical strain, the regulation of NTRK2 transcription during bladder regenerative stimuli is unknown. Therefore, we

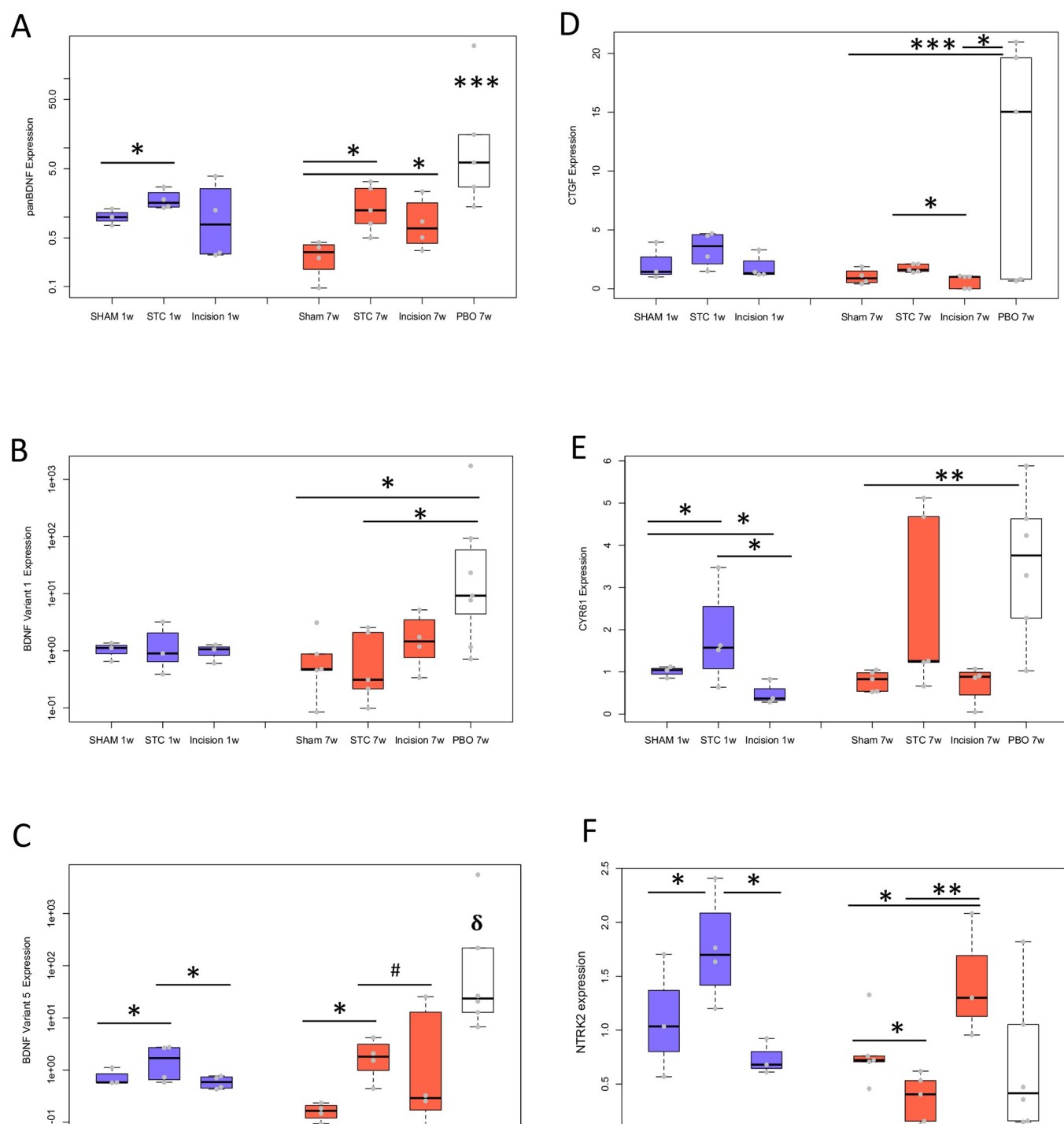

**Fig 3. STC in the bladder leads to increased expression of YAP/WWTR1 target genes. A:** Pan-*BDNF* at 1 and 7 weeks augmented significantly with STC, but not incision at 1week. **B, C**: Isoforms of *BDNF* shows differential regulation in the different injury models. While Variant 1 (exon VI) was not significantly increased during STC **(B),** significant elevation of the Variant 5 (exon IV) isoform was seen during STC (**C**). **D:** *CTGF* mRNA expression was not increased significantly by STC, though incision was significantly lower than STC. **E:** *CYR61* mRNA expression increased during STC at 1 week, with a trend at 7 weeks. Incision alone induced minimal change. **F**: NTRK2 mRNA increased initially at 1 week STC, but decreased by 7 weeks STC. The incision control showed the opposite trend. *, *p* <0.05; **, p<0.01; ***, p<0.005.

quantified the receptor of BDNF *NTRK2* by qPCR after STC or Incision. STC caused an increase of *NTRK2* expression at 1 week, but a decrease at 7 weeks (Fig 3F). In contrast, Incision led to a decrease of *NTRK2* at 1 week, but an increase at 7 weeks.

## STC stimulates nerve fibre regeneration

BDNF, which is known to promote neuronal growth and myelination [36, 37], and neuronal infiltration is increased in response to bladder augmentation [38]. The ability of STC and incision repair to stimulate neuronal infiltration in the bladder was unknown. Therefore, we analyzed the abundance of peripheral neurons in bladders 7 weeks *post*-incision and -STC by immunofluorescence for β3-tubulin (Btub). Smooth muscle was revealed by staining for calponin. The density of nerve fibrils adjacent to smooth muscle was increased by >2-fold, after STC compared to sham, p<0.05, and incision, p<0.01 (Fig 4A and 4C). In the stroma and urothelium, nerve fibre density was higher after STC than sham (p<0.05, Fig 4B and 4C).

## YAP/WWTR1 and BDNF signaling are required for BSMC migration

To assess the potential role of YAP/WWTR1 and BDNF in the migration of BSMC, we quantified wound closure in scratch wound assays. Cell migration into scratch wounds was visualized by live cell microscopy in BSMC monolayers treated with either vehicle, the soluble BDNF receptor NTRK2, or the YAP/WWTR1 inhibitor, VP. Vehicle-treated BSMC migrated quickly, closing much of the wound edge by 12 hours (Fig 5A). Inhibition of YAP/WWTR1 activity using VP reduced the migration of BSMC into the wound by 7 hours, but migration recovered by 12 hours (Fig 5A). Inhibition of BDNF by treatment with soluble NTRK2 (denoted as TRKB) also blocked migration of BSMC (Fig 5A). Furthermore, inhibition of NTRK2 kinase activity by GNF 5837 blocked migration (Fig 5B). In addition, the inhibitor of YAP/WWTR1 signaling, VP, prevented scratch-induced production of BDNF in conditioned media from BSMC, p<0.01 (Fig 5C). In cells treated with VP plus exogenous CTGF or CYR61, no improvement in migration was observed (Fig 5D). In contrast, exogenous BDNF significantly increased early migration in the VP-inhibited cells.

## Yap-dependent changes in SMC differentiation

BDNF effects on SMC marker expression have been described previously [26], but as CYR61 was increased more than CTGF in the STC, we examined the effect of CYR61 vs. CTGF on SMC markers. As CYR61 and CTGF depend on YAP/WWTR1 signaling in BSMC, the contribution of CYR61 and CTGF to SMC marker expression was examined +/- VP. VP significantly increased calponin expression, which was prevented by addition of 10 μM CTGF, but not 10 μM CYR61 (S1 Fig). Interestingly, exogenous Cyr61 increased calponin expression up to levels similar to VP and also increased proliferation above vehicle levels regardless of verteporfin treatment (S1 Fig). Relative to vehicle control, gelatinase activity in conditioned media was increased with exogenous CYR61 protein treatment of SMC, similar to increases in cell number.

## Gene expression of *CYR61* and *NTRK2* correlates with mass and function in bladders subjected to regeneration stimuli

The association between *CYR61* and *NTRK2* gene expression, and bladder function, have not been examined previously. We correlated their expression levels with bladder mass, bladder/body weight ratios and residual volumes (S2A–S2D Fig) following STC at 7 weeks of age. *Cyr61* was positively correlated with residual volumes ($R^2 = 0.33$, S2A Fig). *Ntrk2* expression

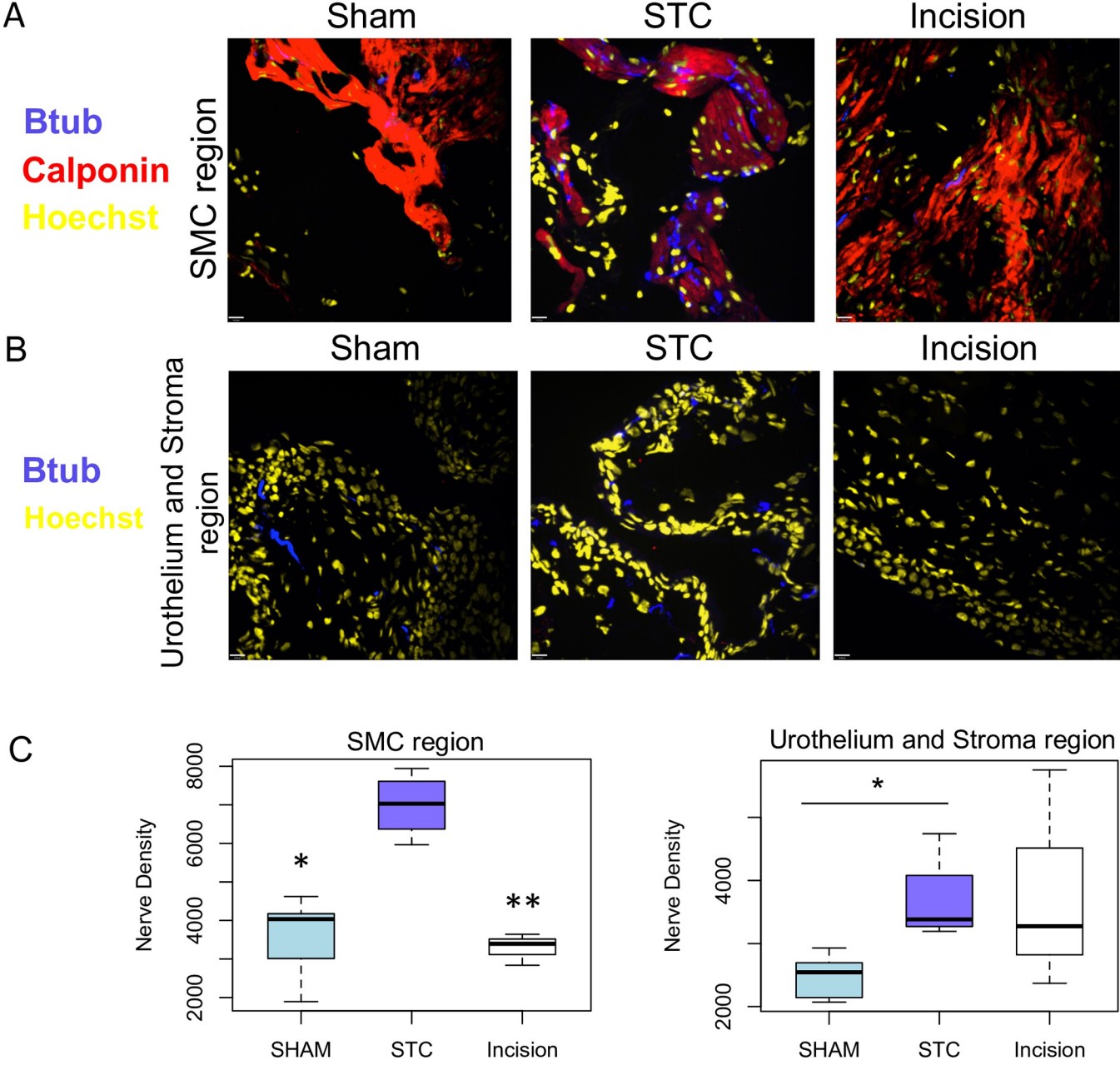

**Fig 4.** Nerve fibre density in regenerating bladders was increased in regions surrounding smooth muscle bundles (**A**) and near the basement membrane and mucosa (**B**). Staining with β3-tubulin antibody (deep blue) demonstrated significantly altered presence of peripheral nerve fibres in the bladders. Smooth muscle was immunostained with calponin (red). (**C**) Quantification of nerve densities in A and B. *, $p < 0.05$.

was positively correlated with bladder mass ($R^2 = 0.46$, S2B and S2C Fig). A comparison of NTRK2 levels from sham, STC and incision with residual volumes yielded a correlation coefficient of $R = -0.86$ and an $R^2 = 0.74$ (S2D Fig).

## Discussion

STC induces the recapitulation of developmental pathways, such as GLI, BMP4 and SHH [16] as well as YAP/WWTR1 which is part of the Hippo/MST1 pathway. The recapitulation of

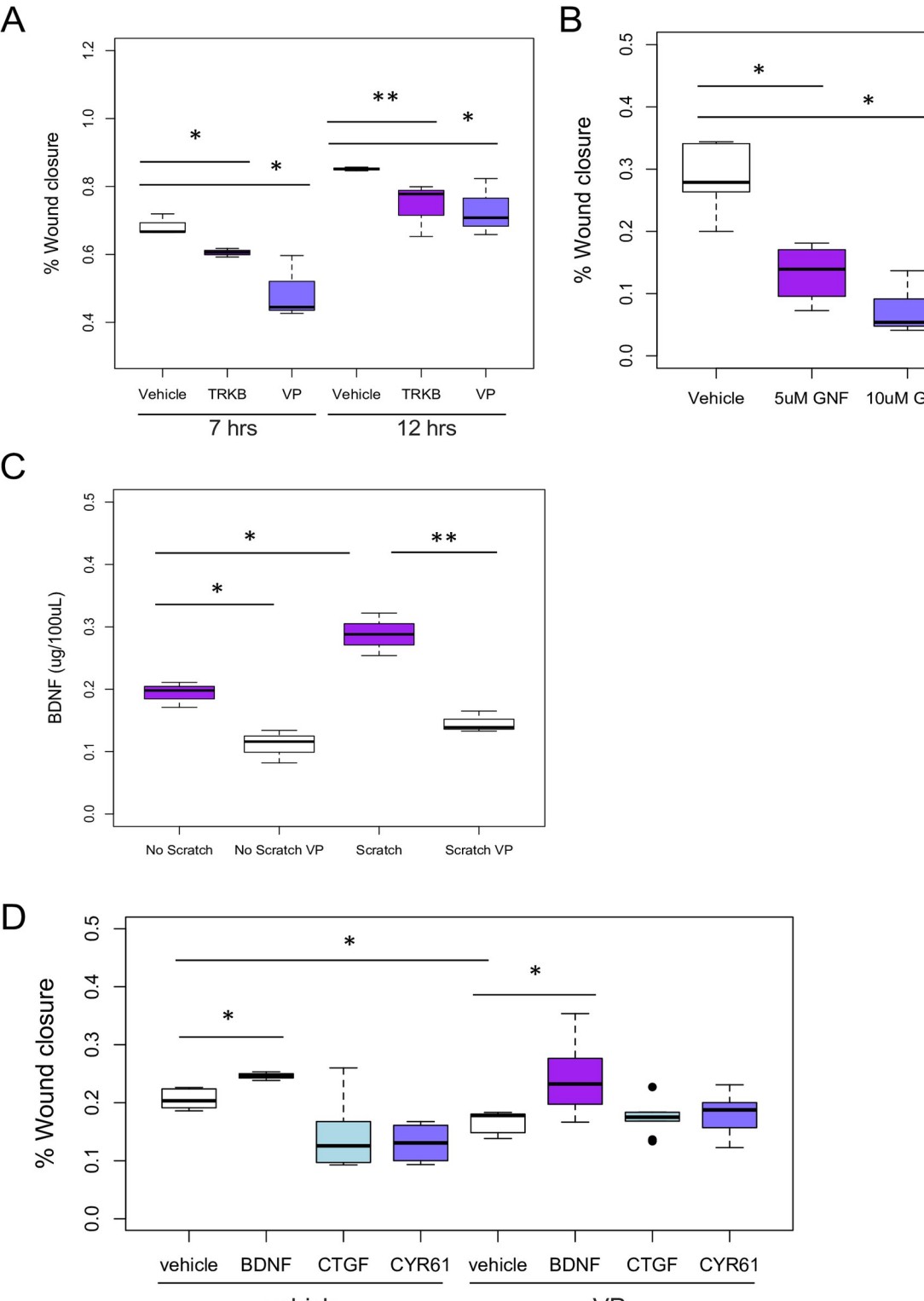

**Fig 5. Migration during scratch wound healing depends on YAP/WWTR1 signaling and BDNF. A: Using** live imaging of the scratch wound edges, migration of bladder smooth muscle cells was assessed over 12 hours, with and without inhibitors (as in Fig 1C, S1 Video). % Closure was calculated by comparing the change in wound area. Wound closure occurred rapidly after initiation, with 68% closure at 7 hours and 85% closure at 12 hours. Soluble BDNF receptor (NTRK2) was able to reduce wound closure at both 7 and 12 hours. Verteporfin (0.1uM) reduced wound closure at 7 hours. **B:** GNF 5837, an inhibitor of the NTRK2 kinase, caused a significant

decrease in bladder SMC migration after scratch wounding, reducing closure by >3-fold. **C:** BDNF protein in scratched BSMC lysates was upregulated during scratch wounds, by ELISA. VP inhibited production of BDNF protein. **D:** Scratch wound was performed with and without VP plus/minus exogenous CTGF, CYR61 and BDNF proteins. BDNF increased migration above vehicle levels, but demonstrated a greater increase relative to the VP-treated scratch wounds which had lower migration levels. $n = 4$ wells were performed for each scratch wound test, with 4–6 fields of view replicates for each n, with student's t-tests $p < 0.05$ considered significant. * $p < 0.05$, **, $p < 0.01$.

developmental responses may derive from a stimulus for the bladder to induce *de novo* growth programs for nerve fibres, urothelium and smooth muscle (Fig 6). Indeed, the muscle, urothelium and other cell types demonstrated increased YAP nuclear localization, consistent with developmental studies showing that YAP is required for muscle growth in bladder [31, 39]. However, all the downstream targets of YAP/WWTR1 have not been defined in the regenerating bladder. Interestingly, Daoud et al (2021) [31] found that muscarinic receptors and smooth muscle markers are dependent on YAP/WWTR1 expression. However, it is unknown, if BDNF, CYR61 and CTGF are intermediate regulators during development. *In vitro*, BDNF was shown to alter the expression of smooth muscle myosin in human BSMC. Also, CYR61 and CTGF, which are highly upregulated during the hypertrophic myopathy model of obstruction both here and elsewhere [26, 28] (Fig 3D and 3E), are known to have effects on matrix and angiogenesis [27, 40]. However, in incision and STC, CTGF remains conspicuously low, suggesting a distinct pattern in the regenerative growth and repair program *in vivo*, which may involve distinct transcriptional co-factors or signaling pathways of YAP/WWTR1.

Distinct physiologic outcomes may arise from differential CTGF expression. CTGF is known to be a crucial factor involved in regulation of TGFbeta and matrix accumulation. We speculated that the unchecked high levels of CTGF during PBO (Fig 3D), in contrast to CTGF absence during bladder regeneration, may lead to differential healing, matrix stiffness and possibly altered SMC cell growth, survival and size. On the other hand, CYR61, which is significantly increased in both STC and PBO, may provide a stimulus for reparative growth and ECM deposition, in the absence of additional signals such as CTGF observed during obstruction (Fig 3D). An understanding of the discrete molecular responses during endogenous bladder regeneration following STC, vs. incision and obstruction, may provide opportunities for therapeutic advancement of bladder engineering and repair. In this study, we contrast YAP/WWTR1 signaling and transcriptional features that distinguish responses of bladder regeneration from both PBO and incisional injury. YAP/WWTR1 signaling is induced during both *in vivo* regeneration and is required for *in vitro* scratch wound healing (Figs 1 and 5). Both CYR61 and CTGF were highly dependent upon YAP/WWTR1, as pharmacologic inhibition of YAP/WWTR1 reduced their expression towards basal levels (Figs 1, 3D and 3E). YAP/WWTR1 activation and downstream gene expression in response to scratch wounding was associated with migration (Fig 1A–1C, S1 Video). Interestingly, while YAP/WWTR1 inhibition led to an increase in differentiation, CTGF addition to VP was able to reverse this differentiation (S1 Fig). Moreover, CYR61 did not induce loss of differentiation, at least in terms of calponin expression (S1 Fig). In addition, WWTR1 and CTGF mRNA rose in response to obstruction, yet fell or did not increase, in response to incision and regeneration, respectively. In summation, YAP and WWTR1 signaling have unique patterns of expression in regeneration as opposed to wounding/incision or obstruction, which may also reflect the expression and roles of CTGF and CYR61.

A discrete number of signaling pathways are known to interact with YAP and WWTR1, including beta-catenin, and mammalian target of rapamycin (mTOR). As we have shown that mTOR is a mechanosensitive pathway that is highly activated during obstruction and mechanical strain [33], it is possible that CTGF and CYR61 may be co-regulated by YAP/WWTR1

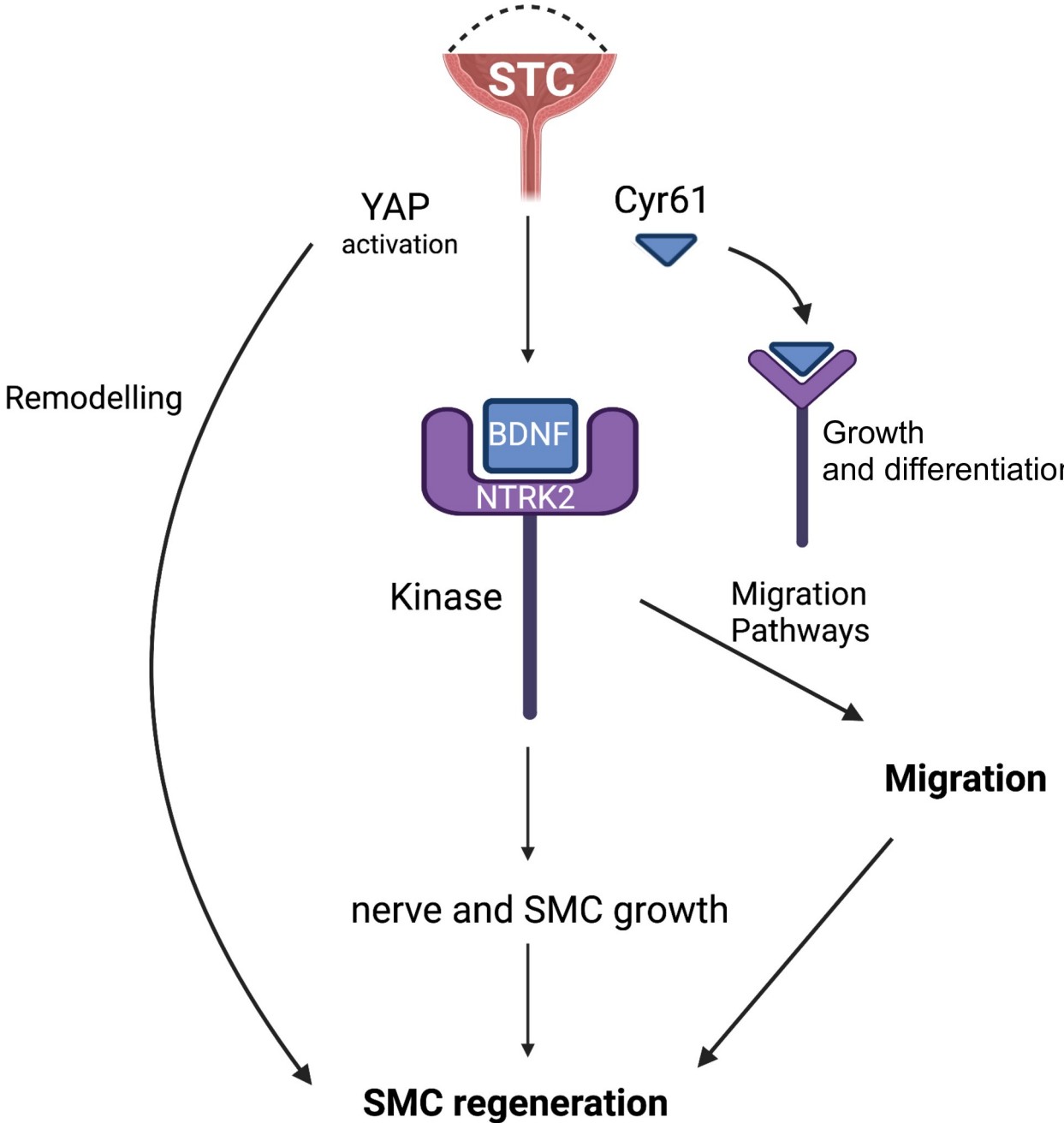

**Fig 6. Schematic of the YAP/WWTR1 signaling pathway responsible for changes after subtotal cystectomy.** Subtotal cystectomy and wound healing induces YAP-dependent *BDNF* and *CYR61* expression. Blocking soluble BDNF decreased migration during scratch wound healing. Pathways to regeneration may also include growth and differentiation pathways that are modulated by BDNF, and CYR61 in smooth muscle cells [33].

and mTOR. In addition, others have shown that CTGF transcription is induced by CREB and NFkB, which have both been implicated in obstructive disease. While CTGF and CYR61 are both considered YAP/WWTR1 targets, the downregulation of WWTR1 during STC may support a role for WWTR1 not YAP in transactivation of the CTGF promoter. It is also possible that WWTR1 may be regulated differentially in distinct bladder cell lineages or more highly regulated by repressive pathways induced during STC.

## Incision and resection from planaria to the bladder

Regeneration as a result of resection has been compared to incisional injury in simpler animal models of regeneration such as planaria and salamanders [41, 42]. In these organisms, incision induces an early wound healing response, as well as proliferation and apoptosis. However, in a later second wave of wound healing, proliferation and apoptosis is only seen in response to resection. Incision alone is resolved early and lacks the positional growth cues that inform the cells that remain after resection that 'tissue is missing', meaning matrix and cells, otherwise known as the "missing tissue response". In a similar fashion here, we have compared partial resection of the bladder to simple incision. We found that the STC (aka partial bladder resection) model shares some expression upregulation of growth regulatory molecules, Cyr61 and Bdnf isoforms, both early and later after resection, which are lost in simple incision (cystotomy) (Fig 3A and 3E). It is interesting to speculate that the regional microenvironment and cell populations, which are stimulated by excision or "missing tissue", may play a role in bladder regeneration. This is supported by clinical studies in which a loss of supratrigonal bladder tissue induced greater regeneration compared to loss of subtrigonal bladder tissue [11]. One possibility is that the "missing tissue response" is not instigated in subtrigonal excision due to distinct regional cellular or matrical cues, or the reported progenitor pool in the trigone is affected [43]. Given the role of CTGF and Cyr61 in cellular and matrix biology, further investigation into the YAP/WWTR1 regulation in the sub vs. supratrigonal models may be of interest.

## Moderate BDNF in regeneration

A newly defined downstream target of YAP/WWTR1, BDNF [26], was also increased in expression during STC *in vivo* and scratch wound *in vitro*. As BDNF has been associated with nerve regeneration, it is significant that BDNF expression showed a coordinated increase in the regenerating bladder. BDNF isoforms are also uniquely differentially expressed between regeneration, incision and hypertrophic myopathy. Both exon IV and VI are highly upregulated during PBO. However, STC does not upregulate exon IV, and shows a much-attenuated increase in exon VI expression (vs. PBO) (Fig 3B and 3C). Incision injury does not induce either of these exons. While the function of BDNF isoforms remains unclear, these data suggest potential specific roles or induction of the isoforms in the context of cell injury responses. BDNF isoforms are known to vary in their cellular and subcellular trafficking patterns. There are some reports of differential post-translational modifications and intracellular localization of particular isoforms, which could lead to differential effects on particular cell types. Whether manipulation of BDNF isoform expression can modulate injury responses from a hypertrophic to a regenerative phenotype, or alter neuronal growth, would be therapeutically advantageous and remains as work for the future.

While initially described as a neurotrophic factor, BDNF may have wider effects on many cell types during injury, cell regeneration, and repair. For example, neuronal regeneration is a crucial part of *in vivo* limb regeneration including production of new mesenchyme in lower amniote models [44]. However, BDNF may also be a part of a smooth muscle (mesenchymal) growth and de-differentiation program induced by various injury models. In our previous work, we demonstrated a direct dose-response role for BDNF in muscle overgrowth and de-differentiation and strong expression in the detrusor muscle tissue [26]. As many YAP/WWTR1-positive cells in the serosal region of the bladder did not stain for differentiated SMC, these could represent either de-differentiated SMC or serosal fibroblasts, which could be resolved through lineage-tracing in future. For example, the observed low levels of CTGF and

BDNF exon IV during STC, but not obstruction, suggests that reduction of these genes during obstruction may promote pathways to regeneration and restoration of bladder function.

BDNF produced by human bladder smooth muscle cells (SMC) in vitro, is important to sustain cell migration, as soluble NTRK2 that binds BDNF to inhibit membrane receptor signaling is able to reduce SMC migration *in vitro*. YAP inhibition also decreased migration consistent with its BDNF expression. The addition of exogenous BDNF to VP treated scratch wounds revealed that BDNF may be a key part of the early migration process. To address whether SMC migration itself occurs in the whole tissue was examined by live cell imaging in *ex vivo* organ culture, which had undergone an *ex vivo* cystectomy. We observed discrete cell movements of individual cells near disorganized regions of the tissue, as well as coordinated moving sheets of cells (Supporting information videos: S2–S5 Videos).

Other signaling pathways that interact with YAP/WWTR1 may be important in this process, such as mTOR, which we have previously found to be important in obstruction [45]. Indeed, when we examined archival cDNA [46] for effects of the mTOR inhibitor rapamycin on obstruction induced BDNF expression, we found that BDNF was highly downregulated by rapamycin, p<0.01 (S3 Fig). The NTRK2 inhibitor GNF 5837 was able to reduce the scratch wound response (Fig 5B). However, GNF can also inhibit PDGF signaling which is known to be mitogenic and pro-migratory to SMC [34]. These data suggest that NTRK2 kinase inhibition sustains migration through continued signaling of the non-kinase truncated NTRK2. A major non-kinase NTRK2 pathway is RhoK, which is known to induce SMC migration [47]. While exogenous RhoK activation induces SMC contraction, inhibition of the NTRK2 kinase in bladder SMC increases contractile function [26], possibly through re-balancing of NTRK2 kinase and non-kinase activities.

## Comparison of residual volumes with gene expression

Interestingly, the NTRK2 non-kinase variant expression is inversely concordant with increases in residual bladder volume, which may be related to bladder contractile function in the context of small residual volumes (Figs 2C and 5A, S2 Fig). Concerning the residual volumes, previous publications showed that mean residual volumes of normal or sham female Sprague-Dawley rats is close to 0.08 mL, but can range up to 0.22 mL [26]. In this publication, we found a lower range of residual volumes. This may be a result of differences in the models, as shams for STC in present experiments were not manipulated near the bladder neck, in contrast to sham operations for obstruction experiments. It should also be noted that in many rodent models minimal but consistent residual volumes are reported in the normal or sham groups, whereas normal clinical results usually do not present with any residual volumes. Nevertheless, results show that the NTRK2 non-kinase variant may have a role in not only SMC migration, but also bladder contractile function.

In terms of regenerated function, it has been noted by some that the regenerated bladder is not completely normal in function, due to hypertrophy and loss of contractile efficiency [48, 49]. However, restoration of bladder function without noticeable histopathologic changes can occur after STC in small rodents [50]. Our study noted that by week 7, STC residual volumes are within the sham range (Fig 2C). In contrast, incised bladders by week 7 exhibited significantly lower residual volumes in comparison to STC, which may be due to the nature and extent of the injury at the respective timepoints (Fig 2C). At 7 weeks, the incision may have resolved and led to increased functionality of the bladder, whereas STC responses may have prevented the reduction of residual volumes. Residual volume changes in both STC and incision controls appear to be concordant with gene expression patterns of NTRK2, BDNF, CYR61 (Figs 2C, 3A, 3E and 3F) which may modify the efficiency of contraction through Rho

kinase, neuronal signaling, matrix changes or SMC de-differentiation. Other possible reasons for functional changes in the bladder include cross-talk from the urothelium (e.g. SHH) that induce a loss of differentiation in the SMC, but these changes may be transitory during early stages after STC. Clearly, bladder function at the terminal stages of regeneration is more clinically relevant while dynamic changes may be expected, as tissue regeneration remains incomplete in the earlier stages. Functional correlations in the three different injury models also benefit the evaluation of potential targets for therapy. Indeed, the positive associations of NTRK2 with bladder/body ratio and bladder mass provide a suggestion that bladder regrowth/hypertrophy may be under partial control of NTRK2 signaling. The positive and negative correlations of CYR61 and NTRK2 with residual volumes suggest that contractile function may also be affected through these genes. Such comparisons make a case to widen our search for similar patterns of expression through genome-wide studies that could open up therapeutic avenues for restoration of bladder function in patients.

## Conclusions

Understanding the endogenous pathways underlying hypertrophic/fibrotic vs. more desirable regenerative bladder injury responses, we may be better able to design optimal conditions for clinical in vivo bladder SMC and urothelial restoration and tissue engineering. These results reveal a bimodal dose-response in terms of tissue phenotype to the expression of YAP/WWTR1 and its downstream transcriptional partners: highly upregulated YAP/WWTR1 target expression (CYR61, BDNF including CTGF) during obstruction, in contrast to attenuated upregulation of CYR61 and BDNF without any rise in CTGF during regeneration. As such, moderate stimulation of the YAP/WWTR1 pathway is possibly a unique mechanistic feature of regeneration. Finally, this example of differential activation in the YAP/WWTR1 pathway provides a model for comparing and contrasting signaling and expression in the regenerating vs. hypertrophic phenotype that could be used to identify factors that may enhance endogenous tissue re-construction as well as engineering of functional bladder tissues.

## Supporting information

**S1 Fig. Immunofluorescent staining for SMC marker calponin in cells on damaged collagen reveals potential for CTGF to cause de-differentiation.** SMC on denatured collagen, were treated with vehicle or the YAP/WWTR1 inhibitor (Verteporfin, 0.1 μM). Addition of CTGF was able to decrease VP-induced differentiation (p<0.05, vs. VP alone), in contrast to CYR61, which did not decrease the effect of VP.
(TIF)

**S2 Fig. Correlation analysis of *CYR61* and NTRK2 expression patterns with residual volumes, bladder mass, and bladder/body mass ratios.** Pearson's correlations were performed with delta or linear values from QPCR for *CYR61* (A) and NTRK2 with or bladder/body mass ratios (B), bladder mass (C) and residual volumes (D).
(TIF)

**S3 Fig. Pan BDNF mRNA expression in sham and obstructed bladders upon rapamycin treatment.** Expression of Pan BDNF is significantly increased in obstructed bladders in comparison to sham, p<0.01. However, Pan BDNF expression is similar between both sham and obstruction bladders upon treatment with rapamycin, an mTOR inhibitor. This animal study was previous published(Schröder et al, 2013, ref 1). Archival cDNA from this study was amplified with panBDNF primers and compared to reference gene results by ddct methods. 1. Schröder A, Kirwan TP, Jiang JX, Aitken KJ, Bägli DJ. Rapamycin Attenuates Bladder

Hypertrophy During Long-Term Outlet Obstruction In Vivo: Tissue, Matrix and Mechanistic Insights. J Urology. 2013 Jun;189(6):2377–84.
(TIF)

**S1 Video. Scratch wound reveals altered movement in bladder SMC.** Bladder SMC either treated with vehicle or verteporfin (VP, an inhibitor of YAP/WWTR1 signaling) were scratch wounded and imaged by time-lapse fluorescent microscopy on a spinning disk confocal microscope (for download see: github S1 Video).
(MOV)

**S2 Video. *Ex vivo* cystectomy reveals altered movement in SMC labelled with AAV6-GFP.** Discrete movement of cells were seen in the labelled region of the bladder. Only discrete regions of the bladder tissue were labelled by the *ex vivo* AAV6-GFP injection. Bladders were imaged by time-lapse fluorescent microscopy on a spinning disk confocal microscope (for download see: github S2 Video).
(MOV)

**S3 Video. *Ex vivo* cystectomy reveals altered movement in SMC labelled with AAV6-GFP.** Another example of discrete movement of myocyte like cells were seen in the labelled region of the bladder. Only discrete regions of the bladder tissue were labelled by the *ex vivo* AAV6-GFP injection. Bladders were imaged by time-lapse fluorescent microscopy on a spinning disk confocal microscope (for download see: github S3 Video).
(MOV)

**S4 Video. *Ex vivo* cystectomy reveals altered movement in SMC labelled with CFSE.** CFSE allowed for general labelling of all cells in the bladder. Here we see the general movement of the *ex vivo* bladder post-subtotal cystectomy (for download see: github S4 Video). Bladders were imaged by time-lapse fluorescent microscopy on a spinning disk confocal microscope.
(MOV)

**S5 Video. *Ex vivo* cystectomy reveals altered movement in SMC labelled with CFSE.** CFSE allowed for general labelling of all cells in the bladder. Here movement of individual cells were seen against a backdrop of general movement with some areas appearing to contract (for download see: github S5 Video). Bladders were imaged by time-lapse fluorescent microscopy on a spinning disk confocal microscope.
(MOV)

## Acknowledgments

We would like to acknowledge the technical help of A. Samiei, the Sickkids Imaging Facility and L. Sung in P. Dirks laboratory.

## Author Contributions

**Conceptualization:** Karen J. Aitken, Martin Sidler, Paul Delgado-Olguin, Darius Bagli.

**Formal analysis:** Karen J. Aitken, Priyank Yadav, Thenuka Thanabalasingam, Prateek Aggarwal, Shing Tai Yip, Nefateri Jeffrey, Jia-Xin Jiang, Aliza Siebenaller.

**Funding acquisition:** Darius Bagli.

**Investigation:** Karen J. Aitken, Priyank Yadav, Martin Sidler, Thenuka Thanabalasingam, Prateek Aggarwal, Shing Tai Yip, Nefateri Jeffrey, Jia-Xin Jiang, Aliza Siebenaller, David Minh Quynh Le.

**Methodology:** Karen J. Aitken, Martin Sidler, Chris Sotiropoulos.

**Project administration:** Darius Bagli.

**Resources:** Paul Delgado-Olguin, Darius Bagli.

**Software:** Karen J. Aitken.

**Supervision:** Paul Delgado-Olguin, Darius Bagli.

**Visualization:** Karen J. Aitken, Thenuka Thanabalasingam, Tabina Ahmed, Prateek Aggarwal, Ryan Huang.

**Writing – original draft:** Karen J. Aitken.

**Writing – review & editing:** Karen J. Aitken, Priyank Yadav, Tabina Ahmed, Darius Bagli.

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
