## [Decision Letter · Decision Letter 0]

30 Mar 2023

PONE-D-22-35632Spontaneous Urinary Bladder Regeneration After Subtotal Cystectomy (STC) Increases YAP/WWTR1

Signaling And Downstream BDNF Expression: Implications For Smooth Muscle Injury Responses.PLOS ONE

Dear Dr. Aitken,

Thank you for submitting your manuscript to PLOS ONE. After careful consideration, we feel that it has merit but does not fully meet PLOS ONE’s publication criteria as it currently stands. Therefore, we invite you to submit a revised version of the manuscript that addresses the points raised during the review process.

We look forward to receiving your revised manuscript.

Kind regards,

Panayiotis Maghsoudlou

Academic Editor

PLOS ONE

Journal Requirements:

3. Please include your animal ethics statement in the Methods section of your manuscript. In the Methods section of your revised manuscript, please include the full name of the animal ethics committee that approved the protocol.

5. Please expand the acronym “CIHR” so that it states the name of your funders in full.

"We gratefully acknowledge the training support of the Strategic Training Program in Regenerative Medicine (TPRM, CIHR) for KJA."

7. Please note that in order to use the direct billing option the corresponding author must be affiliated with the chosen institute. Please either amend your manuscript to change the affiliation or corresponding author, or email us at plosone@plos.org with a request to remove this option.

8. In your Data Availability statement, you have not specified where the minimal data set underlying the results described in your manuscript can be found. PLOS defines a study's minimal data set as the underlying data used to reach the conclusions drawn in the manuscript and any additional data required to replicate the reported study findings in their entirety. All PLOS journals require that the minimal data set be made fully available. For more information about our data policy, please see http://journals.plos.org/plosone/s/data-availability.

9. PLOS requires an ORCID iD for the corresponding author in Editorial Manager on papers submitted after December 6th, 2016. Please ensure that you have an ORCID iD and that it is validated in Editorial Manager. To do this, go to ‘Update my Information’ (in the upper left-hand corner of the main menu), and click on the Fetch/Validate link next to the ORCID field. This will take you to the ORCID site and allow you to create a new iD or authenticate a pre-existing iD in Editorial Manager. Please see the following video for instructions on linking an ORCID iD to your Editorial Manager account: https://www.youtube.com/watch?v=_xcclfuvtxQ

10. Please include your full ethics statement in the ‘Methods’ section of your manuscript file. In your statement, please include the full name of the IRB or ethics committee who approved or waived your study, as well as whether or not you obtained informed written or verbal consent. If consent was waived for your study, please include this information in your statement as well. 

11. We note you have included a table to which you do not refer in the text of your manuscript. Please ensure that you refer to Table 1 in your text; if accepted, production will need this reference to link the reader to the Table.

Reviewers' comments:

Reviewer's Responses to Questions

**Comments to the Author**

1. Is the manuscript technically sound, and do the data support the conclusions?

Reviewer #1: Yes

2. Has the statistical analysis been performed appropriately and rigorously? 

Reviewer #1: Yes

3. Have the authors made all data underlying the findings in their manuscript fully available?

Reviewer #1: Yes

4. Is the manuscript presented in an intelligible fashion and written in standard English?

Reviewer #1: Yes

5. Review Comments to the Author

Reviewer #1: In this manuscript, the authors determined the role of the YAP/WWTR1 pathway and downstream BDNF, CTGF and Cyr61 in the regulation of bladder regeneration. It was found that scratch wound in vitro increased bladder smooth muscle cell migration and expression of BDNF, CTGF and CYR61 in a YAP/WWTR1-dependent manner. Furthermore, bladder regeneration is associated with YAP/WWTR1 signaling and BDNF expression. This manuscript is interesting and well written. I suggest to publish this manuscript after minor revision.

1. Please make sure to define all acronyms the first time they appear in the manuscript. There are some which are never defined, eg. FCS and CFSE.

2. The authors stated “bladder SMC migration is inhibited at the scratch wound edge (Fig. 1C)”. Images from a scratch assay at different time points should be added.

3. How do you explain “Mst1, which inhibits Yap/Wwtr1 activity, is transcriptionally downregulated during scratch wounding, but VP induced a further decrease”.

4. At 1 week, residual volumes in STC were lower than incised bladders, but why residual volumes in incised bladders at 7 weeks significantly decreased when compared to STC (Fig. 2C)?

5. Pan Bdnf was increased in STC vs sham after 1 week (1.68-fold+/-0.53) and 7 weeks (3.6-fold+/-1.1) (Fig. 3A). While the BDNF receptor, NTRK2, increased at 1 week, but decreased at 7 weeks. What's the explanation?

6. The statistical data charts in Figure 1D and E are too small and unclear, which is not convenient for readers to read.

6. PLOS authors have the option to publish the peer review history of their article (what does this mean?). If published, this will include your full peer review and any attached files.

Reviewer #1: No

---

## [Author Response · Author response to Decision Letter 0]

19 May 2023

Response to Review for PONE-D-22-35632

We have now also included the annotated (highlighted in this case, as we utilized google docs to collaborate with authors) and non-annotated versions of the manuscript, as well as the figures that are revised as requested by the reviewer.

1. Please make sure to define all acronyms the first time they appear in the manuscript. There are some which are never defined, eg. FCS and CFSE.

RESPONSE: 

We have revised acryonyms and abbreviations. Abbreviations are now described at their initial use, or are not included if only used once. However, we have left some abbreviations in the title, due to the length of the protein names for WWTR1 (Transcriptional co-activator with a PDZ-binding [WW] domain containing transcription regulator 1) and BDNF (Brain-derived neurotrophic factor). 

2. The authors stated “bladder SMC migration is inhibited at the scratch wound edge (Fig. 1C)”. Images from a scratch assay at different time points should be added.

RESPONSE: 

The original image was supposed to be embedded as a Quicktime movie. We had mentioned this during our initial submission, but unfortunately, the file could not contain a movie easily. We feel that the movies provide the information on the time points, but could revise this if required. Similar to figure S4, Figure 1C has two movies. We have added all of these movies in a zip file containing two pptx files with the movies for Figs.1C and S4, as we were unable to place the movies into pdfs. 

3. How do you explain “Mst1, which inhibits Yap/Wwtr1 activity, is transcriptionally downregulated during scratch wounding, but VP induced a further decrease”.

RESPONSE: 

During scratch wounding, we found that Mst1 was downregulated. This was consistent with the observed increased Yap/Wwtr1 activity during scratch wounding. VP is an inhibitor of Yap/Wwtr1 activity. It is possible that the decrease in their activity leads to a downregulation of Mst1 as a compensation to try to increase the Yap/Wwtr1 activity in the cell. It supports the potential for a negative feedback loop between Yap/Wwtr1 activity and Mst1 transcription, which is disrupted by VP’s inhibition of Yap/Wwtr1 activity.

4. At 1 week, residual volumes in STC were lower than incised bladders, but why residual volumes in incised bladders at 7 weeks significantly decreased when compared to STC (Fig. 2C)?

RESPONSE: 

The incision at 7 weeks may have resolved and become more functional at this timepoint, while in STC ongoing responses may be preventing the reduction of residual volumes. The timecourse for resolution of the responses likely differs due to the nature and extent of the injury, which may also be associated with alterations in signaling and gene expression. This discussion point has been added into the manuscript: “...In contrast, incised bladders by week 7 exhibited significantly lower residual volumes in comparison to STC, which may be due to the nature and extent of the injury at the respective timepoints (Fig 2C). At 7 weeks, the incision may have resolved and led to increased functionality of the bladder, whereas STC responses may have prevented the reduction of residual volumes.”

5. Pan Bdnf was increased in STC vs sham after 1 week (1.68-fold+/-0.53) and 7 weeks (3.6-fold+/-1.1) (Fig. 3A). While the BDNF receptor, NTRK2, increased at 1 week, but decreased at 7 weeks. What's the explanation?

RESPONSE: 

Yes, this is very interesting. It suggests that the particular NTRK2 and BDNF variants may be divergently regulated. Notably, NTRK2 var2 cannot respond to BDNF, as it lacks the kinase region. It appears that during long term STC BDNF expression, this BDNF variant increases as the NTRK non-kinase variant decreases, further increasing the potential for signaling in response to BDNF. We have adjusted the Primer pair designation and clarified this in the text as well.

6. The statistical data charts in Figure 1D and E are too small and unclear, which is not convenient for readers to read.

RESPONSE: 

Figure 1D and 1E have been revised to increase size and clarity.

Editorial Revisions: 

#1. Revisions of style: 

RESPONSE: 

Figure and supplemental figure names were adjusted to PLoSONE standards, as Fig # or S# Fig, respectively. Abbreviations and gene names were adjusted according to Plosone standards, which included HUGO names. 

The exception to this is that we did not change CTGF and CYR61 names to HUGO names CCN2 and CCN1, although both names are mentioned in the introduction. We decided to use the more commonly utilized gene names, CTGF and CYR61, for these two matricellular proteins, as they are utilized most often in the literature, which would make the manuscript more comprehensible. Indeed, a recent PLOS ONE paper utilizes CTGF instead of CCN2, for example: Al-U’datt, et al. (2023). Involvement and possible role of transglutaminases 1 and 2 in mediating fibrotic signalling, collagen cross-linking and cell proliferation in neonatal rat ventricular fibroblasts. PLOS ONE 18(2): e0281320 (https://doi.org/10.1371/journal.pone.0281320). Additionally, the standard nomenclature refers to human genes with capitals and rat names with the first letter as a capital, the rest lowercase. We have utilized this method. 

#2 and 3. Revisions of methodology

“2. To comply with PLOS ONE submissions requirements, in your Methods section, please provide additional information regarding the experiments involving animals and ensure you have included details on (1) methods of sacrifice, (2) methods of anesthesia and/or analgesia, and (3) efforts to alleviate suffering.

3. Please include your animal ethics statement in the Methods section of your manuscript. In the Methods section of your revised manuscript, please include the full name of the animal ethics committee that approved the protocol.”

RESPONSE: 

Methodology details involving animals and the animal care committee are now updated. 

#4 and 6: 

RESPONSE: 

We have harmonized the granting information, but it is important to note that grant numbers are not available for CIHR awards – the PI or fellow name is the only referral. With regard to the funding statement, the funder did not have a direct role in the design, data collection, decision to publish or preparation. While the agency TPRM program approved KJA’s trainee application and trained KJA in regenerative medicine, this specific project was designed and carried out by the supervisors (DJB and PDO), the applicant (in this case KJA) and the accompanying authors. We have revised the statement to include the training aspect of the program: “The funders had no role in study design, data collection and analysis, decision to publish, or preparation of the manuscript, and only played a role in the scientific training of the fellow KJA in regenerative medicine.” 

#5. RESPONSE: The acronym “CIHR” refers to the Canadian Institutes of Health Research. 

#7 and 9: 

RESPONSE: 

The corresponding author is no longer at the Institution where the work was originally performed, but the last author does not have an up-to-date ORCID ID. Both IDs are included here: 

● KJA: 0000-0003-2995-0205

● DB: 0000-0002-4521-3056 

#8: 

RESPONSE: 

The minimal dataset is now being placed into a Github repository under the first author’s auspices: https://github.com/kjaitken/STC_repository. Additional code or instructions for VolocityTM (Perkin Elmer software that is useful for measuring intensities in micrographs) for plotting and analysis is being placed in this repository as well. 

● The values behind the means, standard deviations and other measures reported;

● The values used to build graphs;

● The intensities and other measurements extracted from images for analysis.

Examples of the fluorescent imaging data in support of reported results are provided in the paper itself. We are also in the process of submitting lab protocols on protocols.io as recommended by the editor. 

#10: 

RESPONSE: 

A full “REB” ethics statement might not be required, as this is an animal study which did not have any human component. However, we have updated the animal study information (See #2,3).

#11: 

RESPONSE: 

The Table is now referenced and included immediately after the paragraph where it is first mentioned, as required in the PLOS ONE standards. 

#12: 

RESPONSE: 

We are not aware of any retracted articles in the reference list, but have noted that some of the references may have been flagged for various reasons: 

● Reference 1: Xia et al posted an Author Correction to this article on 27 July 2018.

● Reference 2: had an erratum posted. 

● References 6-10: these are old archival papers that are pertinent to human cystectomy which is rarely performed for non-cancer reasons now. Perhaps these articles appear to be unavailable on your search engine as they are quite old. 

● Reference 18: This review is from the Journal of Tissue Science and Engineering. This journal’s articles appear to be available only through the publisher website after 2012, with most of the articles after 2012 not appearing on pubmed. The publisher website states: “Journal of Tissue Science and Engineering will support authors by posting the published version of articles by NIH grant-holders and European or UK-based biomedical or life sciences grant holders to PubMed Central immediately after publication.“ Though the article is not listed on pubmed, manuscripts submitted to this journal appear to undergo the usual peer review process of an academic journal. 

Finally, we would like to express our appreciation to the editors and reviewer for their review of our piece, and for providing the guidance needed to refine our work. As we believe that this work will benefit the research area of benign urologic disease and basic bladder cell and molecular biology, we want to thank your team for helping us showcase our efforts.

---

## [Editor Report · Decision Letter 1]

1 Jun 2023

Spontaneous urinary bladder regeneration after subtotal cystectomy Increases YAP/WWTR1

signaling And downstream BDNF expression: implications for smooth muscle injury responses.

PONE-D-22-35632R1

Dear Dr. Aitken,

We’re pleased to inform you that your manuscript has been judged scientifically suitable for publication and will be formally accepted for publication once it meets all outstanding technical requirements.

Kind regards,

Panayiotis Maghsoudlou

Academic Editor

PLOS ONE

---

## [Editor Report · Acceptance letter]

17 Jul 2023

PONE-D-22-35632R1 

Spontaneous urinary bladder regeneration after subtotal cystectomy increases YAP/WWTR1 signaling and downstream BDNF expression: implications for smooth muscle injury responses. 

Dear Dr. Aitken:

I'm pleased to inform you that your manuscript has been deemed suitable for publication in PLOS ONE. Congratulations! Your manuscript is now with our production department. 

Kind regards, 

on behalf of

Dr. Panayiotis Maghsoudlou 

Academic Editor

PLOS ONE